# Interstitial 1q Deletion Syndrome: A New Patient with Congenital Diaphragmatic Hernia and Multiple Midline Anomalies

**DOI:** 10.3390/genes16030319

**Published:** 2025-03-07

**Authors:** Gregorio Serra, Rosaria Nardello, Vincenzo Antona, Maria Rita Di Pace, Alessandra Giliberti, Mario Giuffrè, Daniela Mariarosa Morreale, Ettore Piro, Ingrid Anne Mandy Schierz, Maria Sergio, Giuseppina Valenti, Marco Pensabene, Giovanni Corsello

**Affiliations:** Department of Health Promotion, Mother and Child Care, Internal Medicine and Medical Specialties “G. D’Alessandro”, University of Palermo, 90127 Palermo, Italy; gregorio.serra@unipa.it (G.S.); vincenzoantona@virgilio.it (V.A.); mariarita.dipace@unipa.it (M.R.D.P.); gilale98@gmail.com (A.G.); mario.giuffre@unipa.it (M.G.); danielamariarosa.morreale@community.unipa.it (D.M.M.); ettore.piro@unipa.it (E.P.); inschier@tin.it (I.A.M.S.); maria.sergio@unipa.it (M.S.); giusi.valenti.1997@gmail.com (G.V.); marco.pensabene@policlinico.pa.it (M.P.); giovanni.corsello@unipa.it (G.C.)

**Keywords:** CDH, neonatal emergencies, a-CGH, chromosome 1q, contiguous gene syndrome

## Abstract

Background: Interstitial deletions of chromosome 1q are rare, with about 30 cases reported in the literature. The phenotypical features of the affected subjects described so far include microcephaly, pre- and post-natal growth retardation, psychomotor delays, ear anomalies, brachydactyly, in addition to small hands and feet, and rarely a congenital diaphragmatic hernia (CDH). Case presentation: Here, we report on a neonate with CDH, dysmorphic features, and multiple midline anomalies including a cleft palate, in whom an array-comparative genomic hybridization (a-CGH) analysis allowed the identification of an interstitial deletion of the long arm of chromosome 1. Our patient underwent a surgical correction of CDH on the fourth day of life, while that of cleft palate has been planned to be performed at 12 months. Conclusions: The few subjects suffering such rearrangement reported to date, along with the clinical and genetic profile of the present newborn, show that 1q deletions should be considered within the context of the “interstitial 1q deletion syndrome”. Comparing our case with those described in previous studies, the involved genomic regions and the phenotypic traits are partially overlapping, although the clinical picture of the present patient is among the few ones including a congenital diaphragmatic hernia within the phenotypical spectrum. A more extensive comparative analysis of a larger number of patients with similar genetic profiles may allow for a more precise clinical and genomic characterization of this rare syndrome, and for genotype–phenotype correlations.

## 1. Introduction

The congenital diaphragmatic hernia (CDH) results from abnormal diaphragm development, leading to the herniation of abdominal viscera into the thoracic cavity. The CDH is associated with pulmonary hypoplasia and pulmonary hypertension [1]. Its incidence is estimated at approximately 1:4000 live births. In about 5–10% of cases, the CDH is related to genetic syndromes (such as Fryns, Donnai–Barrow, Cornelia de Lange, Beckwith–Wiedemann, Pallister–Killian, and Wolf–Hirschhorn) [2,3,4,5] or other chromosomal anomalies, such as the deletions of chromosome 1q41q42, 8p23.1, and 15q26 [5]. In the recent years, the development of new diagnostic techniques [6,7,8] and therapeutic strategies has significantly improved the prognosis and survival of the affected subjects [1,2]. Therefore, a proper diagnostic assessment, including appropriate genetic testing and individualized treatment is crucial for these patients.

Deletions of the long arm of chromosome 1 are rare and, particularly, the cases of 1q31.1-q31.2 deletion are exceptionally rarely observed. The clinical features of the affected individuals include microcephaly, pre- and post-natal growth retardation, psychomotor delay, ear anomalies, brachydactyly, small hands and feet. Cases reported in the literature so far present some overlapping phenotypic traits, with CDH being documented only in individuals with 1q43-q44 deletions. Here, we report on a neonate with CDH, facial dysmorphisms, and multiple midline anomalies, in whom molecular cytogenetic analysis (array-comparative genomic hybridization, a-CGH) identified a deletion of the long arm of chromosome 1 (1q31.1-q32.1).

## 2. Case Presentation

The proband is a male, second child of healthy and non-consanguineous parents. Family history was unremarkable. During the first trimester of pregnancy, prenatal screening (bi-test) indicated an intermediate risk of trisomy 21, and a high one of fetal growth restriction (FGR). The couple declined any prenatal genetic testing. During the third trimester, polyhydramnios and a left congenital diaphragmatic hernia (CDH) were identified. He was born late preterm, at 36 + 1 weeks of gestation, via an emergency cesarean section due to the failure of labor progression and premature a rupture of membranes. At birth (which occurred at a birthing center in western Sicily, in a nearby province under our Mother and Child Department of the University of Palermo, Palermo, Italy) the Apgar scores were 0, 4, and 6, at 1, 5, and 10 min, respectively. The patient underwent advanced resuscitation maneuvers, including intubation, chest compression and intravenous drugs administration (epinephrine and saline solution, due to persistent bradycardia, considered as <60 beats per minute), and was transferred to the Neonatal Intensive Care Unit (NICU). There, invasive mechanical ventilation, total parenteral nutrition, and cardiovascular support through inotropic drugs were initiated. After the correction of metabolic acidosis, start of antibiotic treatment (also due to increased inflammation markers compatible with early-onset sepsis), and cardio-respiratory stabilization, he was transferred to our Department, in which the Divisions of Pediatric Surgery, Medical Genetics, and Child Neuropsychiatry are present, to continue the diagnostic and therapeutic management. At admission, the anthropometric measures were as follows: weight 3050 g (76th centile; +0.72 standard deviations, SD), length 52 cm (99th centile; +2.21 SD), and occipitofrontal circumference (OFC) 35.5 cm (96th centile; +1.75 SD), according to the Italian INeS Growth Charts [9]. Physical examination revealed a broad and receding forehead, hypertelorism, wide nasal root, bulbous tip, anteverted nares, long and thick philtrum, thin lips, dysplastic, low-set and posteriorly rotated ears with thickened helices, complete (primary and secondary) cleft palate and microretrognathia with glossoptosis (outlining, thus, a Pierre Robin sequence), pectus excavatum and teletelia (Figure 1a–c). Left cryptorchidism and bilateral clinodactyly of the fifth toe completed his clinical profile.

A neurological examination revealed generalized hypotonia, reduced deep tendon reflexes (DTR), diminished primitive reflexes, and decreased responsiveness. Brain ultrasound showed poor gyral development and millimetric cystic lesions within the periventricular white matter. Echocardiography revealed ostium primum type atrial septal defect (0.8 cm), patent foramen ovale and ductus arteriosus, as well as a moderate pulmonary hypertension (pulmonic artery pressure [PAP] values of about 50 mmHg) which subsequently decreased after the medical treatment. Testicular ultrasound identified only the right testis (volume reduced to normal, resulting from the following measures 1 × 0.6 × 0.3 cm, [10,11]) visualized along the inguinal canal. Ophthalmological examination did not reveal any anomalies. In light of the congenital malformations and dysmorphic features found, a molecular cytogenetic analysis (a-CGH) was performed. The genetic test (performed using the platform array-CGH 180K, kit SurePrint G3 Human CGH Microarray Kit, 4×180K, Agilent Technologies, Inc.; the variants have been analyzed through a dedicated software analysis, Agilent Cytogenomics v5.3; the quality of the experiment has been valued as excellent, based on control quality parameters provided by the analysis software [QC metrics] and on the DNA quality) allowed for the identification of a deletion of approximately 12 Mb on the long arm of chromosome 1 (average resolution 50 kb), within the region 1q31.1-q32.1 (GRCh37, proximal and distal breakpoints between 187,956,640 and 199,996,777 bp, respectively), thereafter documented as de novo, and classified as probably pathogenic according to the American College of Medical Genetics and Genomics guidelines [12]. In fact, based on the consulted databases (besides the scientific literature, i.e., UCSC Genome Browser, DatabasE of genomiC variation and Phenotype in Humans using Ensembl Resources [DECIPHER], ClinVar, Human Gene Mutation Database [HGMD], Database of Genomic Variants [DGV], Genome Aggregation Database [gnomAD], the Italian Reference Database, and the Internal Database of the current Laboratory), deletions overlapping the above mentioned one are not reported in the general population (DGV). Moreover, some overlapping deletions are described as probably pathogenic/pathogenic within the reference databases ClinVar (ID VCV001807694.1 and VCV000146285.2) and Decipher (ID 323387 and 341055). On the fourth day of life, the patient underwent a surgical correction of CDH. The defect was large, due to a near-complete agenesis of the left hemidiaphragm, with herniation of the stomach, colon, small intestine, spleen, and left lobe of the liver. After repositioning the viscera into the abdomen, a patch placement was necessary. The following clinical evolution was regular, with enteral nutrition with breast milk being initiated approximately two weeks after surgery, first via a nasogastric tube and then using a special bottle suitable for the cleft palate. After about one month, complete and exclusive bottle feeding was achieved. However, feeding difficulties persisted, associated with frequent regurgitation, for which special anti-regurgitation formula, meal fractionation, and postural therapy have been started, finally leading to beneficial effects and the improvement of symptoms. The patient was then discharged and enrolled in a multidisciplinary follow-up. Neurodevelopmental assessment at 2 months of age (corrected age of 1 month and 13 days) showed poor and repetitive spontaneous movements, generalized hypotonia, but normal passive muscle tone, as well as primitive and deep tendon reflexes. Moreover, a head US evaluation documented no additional and/or evolutive abnormalities. At approximately five months of age (corrected age 4 months; Figure 2a,b), the testes were palpable in both inguinal regions. At the cardiological follow-up evaluation only the persistence of patent foramen ovale was observed. The surgical scar was flat, with no evidence of excessive healing.

At 8 months of age, a chest X-ray revealed a reduction in the opacification of the left hemithorax, with persistence of a parenchymal consolidation in the mid-to-upper lung field, with partial sparing of the marginal zone. The arterial blood analysis evidenced a balanced profile. The neuropsychiatric evaluation disclosed an alert and cooperative patient. Visual engagement, social smiling and response to name were present. The infant was looking for contact with the operator. Pupils were isochoric, isocyclic and normoreactive to light stimuli. The cranial nerves were intact upon exploration. Visual tracking of the object was present both for visual and sound stimuli. The baby recognized familiar faces. Gestures were poor. He manipulated one cube at a time and threw objects on the ground. The exploration was predominantly oral. Generalized hypotonia was still present although mild; a wide excursion at the scarf maneuver was noted along with a popliteal angle between 150 and 160°. Head control, but not trunk one, was nearly completely acquired. In the supine position, he had a tendency towards flexion and external rotation of the lower limbs. Muscular trophism was normal, feet were in varus attitude. Osteo-tendineous reflexes were normally elicitable. Landau reaction was elicitable, Babinski sign was not. Good positioning of the ventral suspension was observed; furthermore, if placed in a prone position he tended to free himself. Good manual grip and lively free motor skills were present. The proband is now 10 months old (9 months of corrected age), and shows a poor growth: weight 7.5 kg (6th centile, −1.57 SD), length 71 cm (33rd centile, −0.44 SD), head circumference 45 cm (50th centile, −0.01 SD), according to World Health Organization growth charts [13]. He does not show any further clinical anomalies, and laboratory tests as well as US multiorgan and neurosensorial evaluations do not evidence other abnormalities to date. Surgical correction of cleft palate is currently planned to be performed at age 12 months.

## 3. Discussion

Deletions of the long arm of chromosome 1 are rare. They are classified into three groups: 1q21-25, 1q25-32, and 1q42-44. The genomic regions involved and the phenotypic features of the reported individuals carrying such rearrangements are partially overlapping with those observed in the present patient [14,15,16]. However, thoraco-abdominal anomalies, which are present in our proband, have been rarely observed in these cases, suggesting that phenotypic expression may vary depending on the extent and type of genes involved in the rearrangement, according to contiguous gene syndromes [17,18,19]. Cases of 1q31.1-q31.2 deletions are exceptionally rarely observed. The first description dates back to 1987 and since then only a few following patients have been reported, whose main clinical, neurological and genomic features are outlined and synthetized in Table 1 [14,15,16,20]. Comparing the cases reported in the literature with our patient (Table 1), similarities can be found both in the phenotype (broad forehead, hypertelorism, broad nasal bridge, bulbous nasal tip, anteverted nares, long and thick philtrum, thin lips, low-set and posteriorly rotated auricles, bilateral clinodactyly of the fifth toe) and in the neurological aspect (hypotonia, feeding problems). The latter, although potentially associated in the baby with a perinatal hypoxic–ischemic injury (IIE), seem more likely linked with his genomic alterations, also in light of lack of prenatal fetal signs and US findings within the follow-up assessments attributable to IIE, and neurological picture and developmental trajectory not suggestive of brain harm due to vascular causes (i.e., absent evolution into increased muscular tone and/or hyperelicitable archaic and osteotendinous reflexes, typically observed in cases of impaired cerebral blood supply). Additionally, psychiatric issues such as aggressiveness, disinhibition, school difficulties, and sleep disturbances have been disclosed, which are not yet evaluable in our patient due to his young age. However, they should be carefully monitored over time, to promptly identify their potential occurrence. In addition, thoraco-abdominal anomalies are also observed in our proband. Actually, these latter anomalies are present in patients with proximal and terminal deletions of chromosome 1. Specifically, congenital diaphragmatic hernia (CDH) is described in 18% of cases of telomeric deletions of chromosome 1, precisely in the q43-q44 region, along with other congenital defects such as cleft palate, clubfoot, and abnormalities of the cerebral gyri [21].

Hemming et al. described a case of 1q43-q44 deletion characterized by microcephaly, neonatal hypotonia, feeding difficulties, gastroesophageal reflux, corpus callosum abnormalities, tetraplegia, bilateral inguinal hernia, and facial dysmorphic features (prominent frontal bossing, downward-slanting palpebral fissures, flattened nasal bridge, thin philtrum, microretrognathia, carp-shaped mouth, and cleft palate) [22]. Some of the phenotypical features observed in the latter patient (carrying a deleted region distal to the current one), as well as in those reported by Hyder et al. and Carter et al. (with overlapping genomic rearrangements and showing short chin and narrow jaw, respectively; see Table 1 [14,15]) belong to the composite picture of Pierre Robin sequence, present also in our proband. Actually, the triad of micrognathia, glossoptosis (with or without cleft palate) and concomitant airway obstruction defines the Pierre Robin sequence (PRS) or malformation [23]. In such congenital defect the primary developmental disturbance is the micrognathia, leading to a retropositioning of the tongue, obstructing the fetal closure of the secondary palatal shelves, eventually causing cleft and neonatal respiratory and feeding problems by mechanical obstruction of the oropharynx [23,24], present in our case. As more than 50 syndromes have been associated with PRS (in addition to single mutant genes, familial occurrence, chromosome abnormalities, vascular disruption, teratogens, and unknown causes), and each of them has a vast heterogenicity in their respective presentations, it often becomes very difficult for clinicians to accurately identify the associated picture [24,25], as occurred in our experience.

Terminal deletions of the long arm of chromosome 1 are also associated with brain anomalies, particularly hypoplasia (or even agenesis in severe cases) of the corpus callosum and reduced representation of cerebral gyri, with the latter being noted also in the present patient [26,27,28]. The literature also shows two cases that, due to their clinical features (congenital diaphragmatic hernia, pulmonary hypoplasia, and dysmorphic facies), were initially misdiagnosed with Fryns Syndrome. Subsequently, array-CGH revealed a microdeletion of 1q41-q42 [29]. A clinical overlap (hypotonia, autism, and facial dysmorphisms such as broad forehead, hypoplastic nasal wings, broad palate, mild retrognathia, everted lower lip, posteriorly rotated ears, pectus excavatum, and clinodactyly of the fifth digit) is also observed in cases of proximal long arm deletions of chromosome 1, specifically of 1q23.3-q24.2 [30]. These cases also show midline anomalies like umbilical hernia, as well as inguinal hernia and renal abnormalities [31,32]. Therefore, the cases reported so far in the literature (both those with overlapping genomic alterations and those with interstitial rearrangements of proximal or distal regions with respect to the one identified in the proband), as well as the clinical and genomic features of the baby, point out and confirm that 1q deletions should be classified within the “interstitial 1q deletion syndrome” [32]. It is likely that the phenotype associated with interstitial 1q microdeletions is influenced not only by the loss of function of the involved genes, but also by other factors such as expression variations in non-deleted genes and/or modifier and/or regulatory genes with effects on other nearby genomic regions. Additionally, epigenetic and environmental factors may play a role in phenotypic variability. This hypothesis is supported by literature data reporting patients who, besides the classic 1q deletion phenotype, exhibited atypical or additional clinical manifestations. In this regard, we refer to the case of an 8-year-old girl with a 1q31-q42 deletion associated with congenital glaucoma [33].

In the deleted region of our patient, several disease-associated genes are present (*CDC73*, *KCNT2*, *CFH*, *CFHR1*, *CFHR5*, *F13B*, *ASPM*, *CRB1*, *PTPRC*), which are associated with various conditions. Pathogenic variants of *KCNT2* are linked with developmental and epileptic encephalopathy; a recent study analyzed 25 patients carrying mutations of this gene with a broad phenotypic spectrum, including intellectual disability, psychomotor/developmental delay, epilepsy, altered muscle tone, and facial dysmorphic features [34]. The literature also documents cases of *KCNT2* variants associated with developmental delay and intellectual disability but without epilepsy [35]. *CDC73* encodes parafibromin, a protein with dual function as both an oncogene and a tumor suppressor, but in cases of haploinsufficiency it may promote tumor development [36]. Its variations are related to a rare syndrome, known as Hyperparathyroidism-Jaw Tumor Syndrome (HPT-JT), characterized by hyperparathyroidism, ossifying fibromas of the mandible and maxilla, increased risk of parathyroid tumors, renal cysts or tumors, thyroid neoplasms, and uterine polyps. *CFHR1* and *CFHR5* are involved in the development of atypical hemolytic uremic syndrome [37] and type II membranous glomerulonephritis [38], respectively. F13B encodes the B subunit of coagulation factor XIII; mutations in this gene can be associated with factor XIII deficiency and consequently with coagulation disorders [39]. Variants of *ASPM* are associated with primary hereditary microcephaly, characterized by microcephaly and intellectual disability of varying severity in the absence of other congenital anomalies [40]. Mutations in *CRB1* are associated with various forms of inherited retinal dystrophies, ranging from milder types such as cone dystrophy to more severe ones such as Leber congenital amaurosis or early-onset retinal dystrophy [41]. In addition to the aforementioned genes, many others are mapped in the 1q31.1-31.2 region (including *TPR* [42] and *BRINP2/3*), whose roles and potential pathogenic implications are not yet fully understood. *BRINP3* is mapped to 1q31.1, and encodes a neuron-specific protein belonging to the BRINP protein family. In murine models where its expression was inhibited, mice showed an altered response to potential danger, particularly a reduction in anxiety, while Brinp2 knockout mice (1q25.2) manifested hyperactivity [43].

A more extensive comparative analysis of a larger number of patients with overlapping genetic profiles will allow for a more precise clinical and genomic characterization of the interstitial 1q deletion syndrome, enabling better genotype-phenotype correlations, also for rare manifestations such as the CDH, which could improve the genomic and clinical characterization of the syndrome.

## 4. Conclusions

We describe a neonate who is among the few cases of 1q31.1-q32.1 deletion reported in the literature to include a CDH. Our observation, thus, expands the database of patients with interstitial 1q deletion syndrome, enhancing its genomic and phenotypic characterization and highlighting the clinical variability of this rare disease. In the diagnostic work-up of patients with CDHs, particularly when facial dysmorphic features and/or multiple congenital defects are present, molecular genetic analyses play a decisive role. Such investigations are crucial also for family counseling. Actually, they enable us to formulate more precise prognosis (including recurrence risk) evaluations, as well as individualized management and follow-up.

## Figures and Tables

**Figure 1 genes-16-00319-f001:**
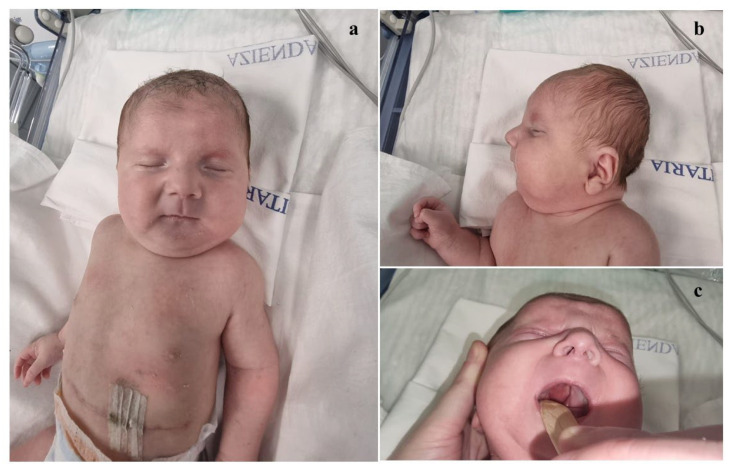
Patient’s phenotype at birth. (**a**) Frontal view: broad and receding forehead, hypertelorism, broad nasal root, bulbous tip, anteverted nares, long and thick philtrum, thin lips along with pectus excavatum and teletelia. (**b**) Lateral view: dysplastic, low-set and posteriorly rotated ears with thickened helices, microretrognathia (within a Pierre Robin sequence). (**c**) Primary and secondary cleft palate.

**Figure 2 genes-16-00319-f002:**
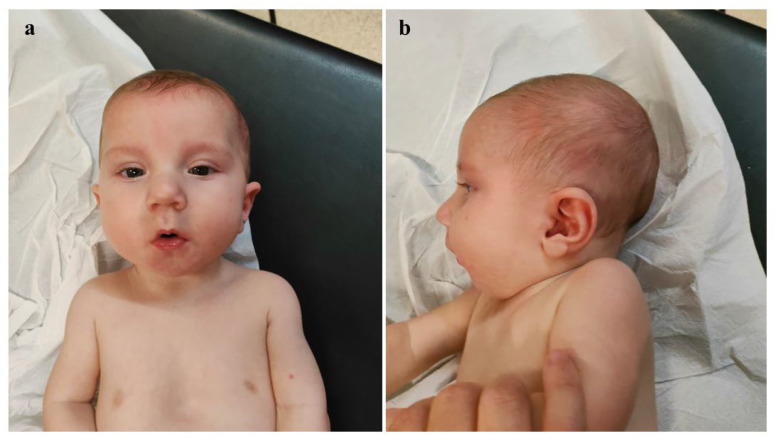
Patient’s phenotype at 5 months. (**a**) Frontal view. (**b**) Lateral view.

**Table 1 genes-16-00319-t001:** Comparison between our patient and the few cases described in the literature with 1q interstitial deletion syndrome and overlapping deleted genomic regions.

Authors	Deletion (Breakpoints, GRCh37)	Dysmorphic Features	Involved Genes	Genetic Test	Cardio-Vascular System	Respiratory System	Neuropsychomotor Profile	Hands and/or Feet Anomalies	Outcome
Milani et al. (2012) [14]	Del 1q 31.1-q32.1 (187,437,627–203,015,924)	broad forehead, laterally sparse eyebrows, slightly downward-slanted palpebral fissures, broad and high nasal bridge, hypoplastic nostrils, long philtrum, thin upper lip and slightly protruding lower lip, retroverted ears	*F13B*, *ASPM*, *CRB1*, *PTPRC*, *PKP1*, *CDC73*, *CACNA 1S*, *TNNT2*	Array-CGH	n.r.	n.r.	Motor, social and cognitive developmental delay		At age 6 years, normal physical growth parameters; mild motor and cognitive developmental delay, hyperactivity and behavioral disorders.
Hyder et al. (2019) [15]	del 1q 31.2-q32.1 (191,590,110–201,139,395)	frontal upsweep, hypertelorism, epicanthic folds, broad nasal bridge, prominent nose, low columella, thin upper lip and everted lower lip, prominent ears, short chin.	*DDX59*, *ASPM*, *CRB1*, *F13B*, *CDC73*, *CFHR5*, *CACNA1S*, *UCHL5*, *TROVE2*, *B3GALT*, *ZBTB41*, *CAMSA2*, *KIF21B*, *TMEM9*	Array-CGH	n.r.	n.r.	At birth, hypotonia and feeding difficulties. Subsequently, developmental delay, hyperactivity, aggression, disinhibition, and sleep disturbances.	Clinodactyly, single palmar crease on left hand, tapering fingers, deep-set small nails.	At 31 years, head circumference 57.4 cm (50th–75th centile), height 174.8 cm (25th–50th centile) and weight 140.6 kg (>99th centile). Downslanting palpebral fissures, broad nasal bridge, low-hanging columella, thin upper lip, thick lower lip, deep-set small nails and tapering fingers. He currently lives independently in a flat with supported living. His main difficulties are with arithmetic and finances, but his memory is good, and he is able to read and write independently
Carter et al. (2016) [16]	Del. 1q32.1 (199,985,888–203,690,832)	Long face, narrow jaw, down-slanted palpebral fissures, highly arched eyebrows, low-set ears, thick lower lip.	*KDM5B*, *NAV1*, *KIF21B*, *GPR37L*, *SYT2*	Array-CGH	n.r.	n.r.	Global developmental delay, social skills and language difficulties, reduced IQ. Generalized hypotonia and decreased deep tendon reflexes	Bilateral clinodactyly of the fifth finger and proximal positioning of the thumb	Neuropsychological evaluation at 7 years of age: full scale IQ of about 50 (Woodcock-Johnson Tests of Cognitive Abilities), difficulties in visual-motor coordination. Significant difficulties with receptive and expressive language; slow improvement in language acquisition. At 10 years, he requires special education and support in everyday life
Our patient	Del.1q31.1-q32.1 (187,95,640–199,996,777)	Broad and sloping forehead, hypertelorism, wide nasal bridge, bulbous nasal tip, anteverted nares, long and thick philtrum, thin lips, dysplastic auricles with thickened helices, low-set and posteriorly rotated ears, complete cleft palate, microretrognathia.	*CDC73*, *KCNT2*, *CFH*, *CFHR1*, *CFHR5*, *F13B*, *ASPM*, *CRB1*, *PTPRC*	Array-CGH	ostium primum-type atrial septal defect	Congenital diaphragmatic hernia	Generalized hypotonia, diminished deep tendon reflexes, reduced primitive reflexes and reactivity, poor cortical gyration, and millimeter-sized cystic lesions in the periventricular white matter	clinodactyly of the fifth finger	At age 8 months, generalized mild hypotonia. Nearly completely acquired head control, not that of the trunk. In the supine position, tendency towards flexion and external rotation of the lower limbs. Normal muscular trophism, feet in varus attitude. Good manual grip and lively free motor skills. Normally elicitable osteo-tendineous reflexes and Landau reaction, not the Babinski sign.

n.r. = not reported.

## Data Availability

The original contributions presented in this study are included in the article. Further inquiries can be directed to the corresponding author.

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
