# Peer review of "Interstitial 1q Deletion Syndrome: A New Patient with Congenital Diaphragmatic Hernia and Multiple Midline Anomalies"

_genes, 2025, doi:10.3390/genes16030319_

Round 1
Reviewer 1 Report
Comments and Suggestions for Authors
Serra and colleagues submit the case report describing a neonate with congenital diaphragmatic hernia as part of the syndrome that features multiple additional anomalies and developmental sequelae, and which are associated with an interstitial deletion of the long arm of chromosome 1. The novelty of the association of this deletion with CDH, as well as the description of associated anomalies and functional outcomes until late infancy, make this report worth publishing. However, the manuscript includes many inconsequential details which detract from its readability.
Some suggestions for making this paper more useful to readers are listed below.
The Abstract is too long, both in the background and the case presentation sections. In particular, the timing of the surgical correction of the CDH, and plans for the correction of the cleft palate are necessary here. However, there is sufficient detail for readers who may only peruse the abstract and skip the body text.
In the body text, the Background and the references included are adequate.
The case presentation is missing a couple of potentially useful details. The progression of Apgar scores does not make complete sense, particularly the Apgar of 0 at 1 minute. If this was truly the case, the baby would be at risk for hypoxic ischemic encephalopathy and possible long-term neurodevelopmental sequelae, which might be confounders unrelated to the syndrome. Therefore, it would be important to include a description of the fetal heart rate just prior to delivery, as well as cord blood gases, some comment of whether there was clinical evidence of acute, progressive neurologic dysfunction suggestive of HIE (or whether this was absent). I suspect that the Apgar of 0 at 1 minute is likely unreliable, but the authors should comment on this in the Discussion section.
The postoperative clinical course between lines 120 and 132 is much too detailed, and irrelevant for the aim of the paper, which attempts to establish clinical and genomic correlations. This contents could be effectively summarized in a couple of sentences.
The Discussion contains an appropriate amount of detail comparing this case with other 1q deletion syndromes, but again there is unnecessary verbosity, as well as inappropriate repetition. If a table of prior cases is to be used, there should be no duplication of information between the table and the text; the table might contain key comparison points, and the text might comment on these key comparisons and add other necessary details.
In the Conclusions, the first sentence is entirely unnecessary as this stage of the paper, and there is no reason to include both the name of the condition and the abbreviation, either. This section could begin with “We describe a neonate who is among the few cases…”
Finally, it is unclear how the authors were allowed to obtain consent from the patient's parents for publication without having this cleared by an institutional review board. I would expect that the IRB would need to approve the consent form to be signed by the parents.
Author Response
Serra and colleagues submit the case report describing a neonate with congenital diaphragmatic hernia as part of the syndrome that features multiple additional anomalies and developmental sequelae, and which are associated with an interstitial deletion of the long arm of chromosome 1. The novelty of the association of this deletion with CDH, as well as the description of associated anomalies and functional outcomes until late infancy, make this report worth publishing. However, the manuscript includes many inconsequential details which detract from its readability.
Some suggestions for making this paper more useful to readers are listed below.
The Abstract is too long, both in the background and the case presentation sections. In particular, the timing of the surgical correction of the CDH, and plans for the correction of the cleft palate are necessary here. However, there is sufficient detail for readers who may only peruse the abstract and skip the body text.
Response: The abstract has been shortened both in the background and case presentation, according to the reviewer’s suggestions. The timing of the surgical correction of the CDH, and that for the cleft palate have been properly included.
In the body text, the Background and the references included are adequate.
The case presentation is missing a couple of potentially useful details. The progression of Apgar scores does not make complete sense, particularly the Apgar of 0 at 1 minute. If this was truly the case, the baby would be at risk for hypoxic ischemic encephalopathy and possible long-term neurodevelopmental sequelae, which might be confounders unrelated to the syndrome. Therefore, it would be important to include a description of the fetal heart rate just prior to delivery, as well as cord blood gases, some comment of whether there was clinical evidence of acute, progressive neurologic dysfunction suggestive of HIE (or whether this was absent). I suspect that the Apgar of 0 at 1 minute is likely unreliable, but the authors should comment on this in the Discussion section.
Response: This point has been addressed in the discussion as well as in the case presentation. Detailed data on Apgar score have been verified, and they may be considered reliable since they are provided within the transfer report from the birthing center, including heart rate as well as EGA profile. Despite the right consideration and suspicion raised by reviewers about a potential co-occurrent mechanism responsible for IIE, however there are several aspects suggesting a more probable neurological impairment due to genetic cause than an acquired one: 1) the absence of prenatal cardiotocographic abnormalities; 2) following serial brain US evaluations not compatible with an evolutive picture of IIE; 3) above all the careful clinical assessment made both by neonatologists first and then by child neupsychiatrists which put in evidence a clinical trajectory not suggestive for cerebral damage due to vascular injury (ie hypotonia tending to improve in the following weeks/months, normal OT and archaic reflexes, and absent evolution into hypertonia and/or paresis). All these aspects underline and corroborate how the neurological profile of the proband seems more likely to be associated to his genomic alteration than to the effects due to a less plausible, although possible, brain perinatal damage (see line 56, 157-163).
The postoperative clinical course between lines 120 and 132 is much too detailed, and irrelevant for the aim of the paper, which attempts to establish clinical and genomic correlations. This contents could be effectively summarized in a couple of sentences.
Response: This part of the clinical course has been adequately shortened and synthetized (see lines 104-111)
The Discussion contains an appropriate amount of detail comparing this case with other 1q deletion syndromes, but again there is unnecessary verbosity, as well as inappropriate repetition. If a table of prior cases is to be used, there should be no duplication of information between the table and the text; the table might contain key comparison points, and the text might comment on these key comparisons and add other necessary details.
Response: This section has been significantly shortened and condensed, avoiding inappropriate repetition or duplication with the table, highlighting key comparison points, and including only a few necessary details (see lines 151-153).
In the Conclusions, the first sentence is entirely unnecessary as this stage of the paper, and there is no reason to include both the name of the condition and the abbreviation, either. This section could begin with “We describe a neonate who is among the few cases…”
Response: The conclusion has been accordingly modified (see line 247).
Finally, it is unclear how the authors were allowed to obtain consent from the patient's parents for publication without having this cleared by an institutional review board. I would expect that the IRB would need to approve the consent form to be signed by the parents.
Response: The approval of IRB has been requested to the Bioethics Committee which our University Hospital belongs to (Palermo 1). The request was duly submitted, however approval can only be provided after the committee meeting, scheduled for the next February 28th. In the meantime, the form used for the authorization of legal representatives to publish images and/or videos that has already been authorized by our Department, signed by the parents of our patient who gave consent and that in similar reports might be enough, is provided in attachment as supplementary file.
We thank all the reviewers for the opportunity to clarify some critical points of the manuscript, improving its clarity and overall quality. Hoping that the changes made will meet with approval, we take this time to send our warmest regards
Reviewer 2 Report
Comments and Suggestions for Authors
Dear Authors,
An interesting mansucript, significant in the field and gives overview of the problem. Just one thing, - please, mention an Ethical Committee permisssion (year, code, issue office) and Consent form of the parents as child is recognizable and this info is mandatory in this case!
Author Response
An interesting manuscript, significant in the field and gives overview of the problem. Just one thing, - please, mention an Ethical Committee permisssion (year, code, issue office) and Consent form of the parents as child is recognizable and this info is mandatory in this case!
Response: The approval of IRB has been requested to the Bioethics Committee which our University Hospital belongs to (Palermo 1). The request was duly submitted, however approval can only be provided after the committee meeting, scheduled for the next February 28th. In the meantime, the form used for the authorization of legal representatives to publish images and/or videos that has already been authorized by our Department, signed by the parents of our patient who gave consent and that in similar reports might be enough, is provided in attachment as supplementary file.
We thank all the reviewers for the opportunity to clarify some critical points of the manuscript, improving its clarity and overall quality. Hoping that the changes made will meet with approval, we take this time to send our warmest regards
Reviewer 3 Report
Comments and Suggestions for Authors
The manuscript entitled” Interstitial 1q Deletion Syndrome: A New Patient with Congenital Diaphragmatic Hernia and Multiple Midline Anomalies” was reviewed. This is a well-described rare case. I have only I comments to authors to respond to:
The lateral photographic view showed that the patient has underdeveloped jaw therefore probably this clinical phenotype also belongs to Pierre Robin sequence (also known as Pierre Robin syndrome or Pierre Robin malformation). Please make a new paragraph discussing this aspect.
Author Response
The lateral photographic view showed that the patient has underdeveloped jaw therefore probably this clinical phenotype also belongs to Pierre Robin sequence (also known as Pierre Robin syndrome or Pierre Robin malformation). Please make a new paragraph discussing this aspect.
Response: This aspect has been discussed both in the case presentation and discussion (see lines 75-76, 82, 180-194).
We thank all the reviewers for the opportunity to clarify some critical points of the manuscript, improving its clarity and overall quality. Hoping that the changes made will meet with approval, we take this time to send our warmest regards
Reviewer 4 Report
Comments and Suggestions for Authors
In this article, authors clearly describe the case of an infant with congenital diaphragmatic hernia and chromosome I q deletion, suggesting the introduction of “interstitial 1 q deletion syndrome” word. Two minor issues:
-authors wrote: “The study did not require ethical approval”. I think it did require ethical approval.
-In References, authors have 19 titles written before 2020 (old), out of 39 total titles. They are all well chosen still, some of the old ones could be replaced with some newer ones.
Author Response
Two minor issues:
-authors wrote: “The study did not require ethical approval”. I think it did require ethical approval.
-In References, authors have 19 titles written before 2020 (old), out of 39 total titles. They are all well chosen still, some of the old ones could be replaced with some newer ones.
Response: The approval of IRB has been requested to the Bioethics Committee which our University Hospital belongs to (Palermo 1). The request was duly submitted, however approval can only be provided after the committee meeting, scheduled for the next February 28th. In the meantime, the form used for the authorization of legal representatives to publish images and/or videos that has already been authorized by our Department, signed by the parents of our patient who gave consent and that in similar published reports might be enough, is provided in attachment as supplementary file.
6 updated bibliographic entries, all of them published from 2020 onwards, have been included: see references 6, 25-27 and 29-30. Conversely, self-citations have been shortened, and currently correspond to around 15% of the total as indicated by the Editorial Office.
We thank all the reviewers for the opportunity to clarify some critical points of the manuscript, improving its clarity and overall quality. Hoping that the changes made will meet with approval, we take this time to send our warmest regards